# Serum anti-erythropoietin antibodies among pregnant women with Plasmodium falciparum malaria and anaemia: A case-control study in northern Ghana

Charles Nkansah[1,2]*, Simon Bannison Bani[3], Kofi Mensah[1,2], Samuel Kwasi Appiah[1,2], Felix Osei-Boakye[4], Gabriel Abbam[1], Samira Daud[1], Eugene Mensah Agyare[3], Peace Esenam Agbadza[3], Charles Angnataa Derigubah[5], Dorcas Serwaa[6], Francis Atoroba Apodola[7], Yeduah Quansah[3], Rahama Issah[3], Samuel Yennuloom Dindiok[3], Felix Ejike Chukwurah[2]

**1** Department of Haematology, School of Allied Health Sciences, University for Development Studies, Tamale, Ghana, **2** Department of Medical Laboratory Sciences, Faculty of Health Science and Technology, Ebonyi State University, Abakaliki, Nigeria, **3** Department of Biomedical Laboratory Sciences, School of Allied Health Sciences, University for Development Studies, Tamale, Ghana, **4** Department of Medical Laboratory Technology, Faculty of Applied Science and Technology, Sunyani Technical University, Sunyani, Ghana, **5** Department of Medical Laboratory Technology, School of Applied Science and Arts, Bolgatanga Technical University, Bolgatanga, Ghana, **6** Department of Obstetrics and Gynecology, C4C Homeopathic Medical College, Accra, Ghana, **7** Department of Medical Diagnostics, College of Nursing and Allied Health Sciences, Nalerigu, Ghana

* cnkansah86@yahoo.com

**Data Availability Statement:** The raw data for the paper has been deposited at flowrepository.org (FR-FCM-Z5RW).

## Abstract

### Background

Anaemia in pregnancy is common in underdeveloped countries, and malaria remains the predominant cause of the condition in Ghana. Anti-erythropoietin (anti-EPO) antibody production may be implicated in the pathogenesis of *Plasmodium falciparum* malaria-related anaemia in pregnancy. This study ascertained the prevalence of anti-EPO antibody production and evaluated the antibodies' relationship with *Plasmodium falciparum* malaria and malaria-related anaemia in pregnancy.

### Methods

This hospital-based case-control study recruited a total of 85 pregnant women (55 with *Plasmodium falciparum* malaria and 30 controls without malaria). Venous blood was taken from participants for thick and thin blood films for malaria parasite microscopy. Complete blood count (CBC) analyses were done using an automated haematology analyzer. Sandwich enzyme-linked immunosorbent assay (ELISA) was used to assess serum erythropoietin (EPO) levels and anti-EPO antibodies. Data were analyzed using IBM SPSS version 22.0.

### Results

Haemoglobin ($p<0.001$), RBC ($p<0.001$), HCT ($p = 0.006$) and platelet ($p<0.001$) were significantly lower among pregnant women infected with *Plasmodium falciparum*. Of the 85

**Funding:** The authors received no specific funding for this work.

**Competing interests:** The authors have declared that no competing interests exist.

participants, five (5.9%) had anti-EPO antibodies in their sera, and the prevalence of anti-EPO antibody production among the *Plasmodium falciparum-infected* pregnant women was 9.1%. *Plasmodium falciparum*-infected pregnant women with anti-EPO antibodies had lower Hb ($p<0.001$), RBC ($p<0.001$), and HCT ($p<0.001$), but higher EPO levels ($p<0.001$). Younger age ($p = 0.013$) and high parasite density ($p = 0.004$) were significantly associated with *Plasmodium falciparum*-related anti-EPO antibodies production in pregnancy. Also, younger age ($p = 0.039$) and anti-EPO antibody production ($p = 0.012$) related to the development of *Plasmodium falciparum* malaria anaemia in pregnancy.

## Conclusion

The prevalence of anti-EPO antibodies among pregnant women with *Plasmodium falciparum* malaria was high. *Plasmodium falciparum* parasite density and younger age could stimulate the production of anti-EPO antibodies, and the antibodies may contribute to the development of malarial anaemia in pregnancy. Screening for anti-EPO antibodies should be considered in pregnant women with *P. falciparum* malaria.

## Introduction

Anaemia remains the most prevalent haematological stress in pregnancy and may cause life-threatening complications to foetal and maternal lives. Gestational anaemia is a global problem but the situation is worse in developing countries where malaria is endemic [1]. The development of anaemia in pregnancy is multifaceted, ranging from physiological to pathological aetiologies.

The physiological changes that are seen in pregnancy are mainly attributed to hormonal changes, as well as the astronomical increase in plasma volume (40–45%) relative to the 20–25% increase in red blood cell mass, leading to a reduction in haemoglobin (Hb) concentration [2]. Pathologically, malaria (predominantly *Plasmodium falciparum* malaria) remains the leading cause of gestational anaemia in underdeveloped countries, and contributes significantly to maternal and infant morbidity and mortality [3–5]. In Ghana, the prevalence of *P. falciparum* malaria in pregnancy is 8.9% [6], and yearly, as many as 10,000 maternal deaths attributable to malaria-related anaemia are recorded in sub-Saharan Africa [7]. Malaria in pregnancy may be associated with adverse complications such as maternal and foetal anaemia [6], intrauterine growth retardation, infant mortality, low birth weight, stillbirth and other congenital defects [8]. The pathophysiology of *P. falciparum* malaria-related anaemia in pregnancy is still complicated, and may occur through the suppression of erythropoiesis, placental sequestration of parasitized and non-parasitized erythrocytes, haemolysis of both infected and non-infected erythrocytes, inflammatory mediators such as tumour necrosis factor-alpha (TNF-α), interleukin (IL)-1 and IL-6, and direct interaction with parasite variant surface antigens [4, 9–11].

Immediate erythropoietic response is required to avert complications that may occur during *P. falciparum* malarial anaemia in pregnancy [12]. Upregulation of EPO in malaria-related anaemia is essential for effective bone marrow compensatory response to correct the anaemia. Previous studies found elevated serum erythropoietin levels in patients who presented with severe malaria-related anaemia in India [13], Kenya [14], Sudan [15] and Thailand [16], but could not significantly correct the anaemia. An earlier study involving 18-65-year-old adults also observed defective haemopoietic response vindicated by reduced erythropoietin production as well as reticulocyte response in patients with acute *P. falciparum* malaria [11].

An autologous serum factor that could repress the expected haemopoietic response has been proposed as a plausible cause for the development of the associated anaemia in such patients [17]. The plausible serum factor proposed to have been responsible for the suppressed erythropoiesis seen in malaria-related anaemia remains unclear. Patients who were administered recombinant human erythropoietin developed anti-EPO antibodies, and eventually suffered from pure red cell aplasia [18]. The anti-EPO antibodies may have cross-reacted with endogenous erythropoietin, suppressing its biological functions and subsequently worsening the patients' anaemia. Again, anti-EPO antibodies have been identified in human immunodeficiency virus/acquired immunodeficiency syndrome (HIV/AIDS) and autoimmune diseases like systemic lupus erythematosus [19]. External factors including *P. falciparum* malaria may induce a pro-inflammatory environment that promotes the activation of auto-reactive lymphocytes; this could result in the production of antibodies such as anti-EPO which may interfere with the biological functions of EPO and cause anaemia [9, 11].

Even though a recent study in Ghana involving children aged 1–10 years presenting with malaria did not find any significant association between anti-erythropoietin antibodies and malaria-related anaemia [20], this study was limited to children and could not account for the changes that may occur in pregnancy. The possible changes during pregnancy, and especially in the presence of infections may trigger immune responses including induction of autoimmune-mediated antibodies such as anti-EPO which could contribute to the development of anaemia.

Regardless of the possible significant influence of anti-EPO antibodies in the development of anaemia, no known study, to the best of our knowledge, has assessed anti-erythropoietin antibodies in pregnancy. Therefore, this study assessed the production of anti-EPO antibodies and their association with anaemia in *P. falciparum*-infected pregnant women. Assessment of anti-EPO antibodies in pregnancy may contribute to the understanding of the pathophysiology of *P. falciparum* malarial anaemia, and influence treatment guidelines especially in severe anaemia.

## Materials and methods

### Study site/Design

This was a hospital-based case-control study from January-August, 2022 at the Antenatal Clinic of the Tamale Teaching Hospital (TTH), Northern Region, Ghana. TTH is located in the Tamale Metropolis and serves as a tertiary-level referral facility for the five northern regions: Savannah, Northern, North-East, Upper West and Upper East regions. Tamale has a population of approximately 371,351 with the majority being farmers. The capital of the Northern region, Tamale, is located at a longitude of 0.8235˚W and latitude of 9.3930˚N. The digital address of the hospital is NT-0101-5777. The hospital has an antenatal clinic (ANC) which receives on average, a hundred (100) pregnant women each day [21].

### Study population

Pregnant women attending ANC at the TTH constituted the study population.

### Sample size determination

The minimum required sample size was obtained by employing Kelsey's formula:

$$N_{cases-Kelsey} = \left[\frac{r+1}{r}\right] \frac{P(1-P)\left(Z_{\frac{\alpha}{2}} + Z_{\beta}\right)^2}{(p1-p2)^2}, \text{ and } P = \left[\frac{p1 + (r \, X \, p2)}{r+1}\right]$$

Where "*N*"$_{cases\text{-}Kelsey}$ is the required sample size for the *P. falciparum* mal4aria group.

"r" is the ratio of the *P. falciparum* malaria group to the group without malaria, which is 2:1 in this study.

$Z_{\frac{\alpha}{2}}$ represents the critical value of the normal dispersion at α/2 (for this study at a confidence level of 95%, "α" is 0.05 and the critical value is 1.96).

$Z_{\beta}$ represents the critical value of the normal distribution at β (this study used a power of 80%, "β" is 0.2 and the critical value is 0.84).

p1 represents the prevalence of *P. falciparum* malaria in pregnancy, 8.9% [6].

p2 is the prevalence of anaemia in pregnancy, 37.3% [6].

p1-p2 is the smallest difference in proportions that is clinically important.

The minimum number of participants with *P. falciparum* malaria required for this study was 31, with the corresponding number of participants without malaria at 16.

However, to increase the statistical power, this study employed 85 subjects: 55 pregnant women with *P. falciparum* malaria and 30 controls (pregnant women without malaria).

## Inclusion/Exclusion criteria

Participants for the study were pregnant women in all stages of pregnancy, presenting with *P. falciparum* malaria and anaemia. *P. falciparum*-infected pregnant women with comorbidities (such as infections and other haematological disorders) were excluded. Pregnant women's history of HIV, hepatitis B surface antigen (HBsAg), hepatitis C virus (HCV), Syphilis, Sickling test, Hb phenotype, glucose-6-phosphate dehydrogenase (G6PD) test, routine examinations of stool and urine, as well as other tests had been recorded in the ANC register.

## Sources of data

Participants' socio-demographic and clinical characteristics were obtained by a structured questionnaire and a review of the hospital's ANC records respectively. Also, blood samples were taken from the participants for laboratory assay.

## Sample collection and processing

Six (6) mL of blood samples from participants were collected aseptically: 3 mL into dipotassium ethylenediaminetetraacetic acid (K$_2$EDTA) tubes for CBC and blood film for malaria parasites; 3 mL into serum-separator tubes for analysis of serum EPO levels and anti-erythropoietin antibodies. The tubes were code-labelled to ensure the anonymity of participants. Blood samples in the serum-separator tubes after clotting were centrifuged to separate the serum from the cells. The serum components were aliquoted into Eppendorf tubes and stored at -20˚C until analyzed for EPO and anti-EPO antibodies using Human EPO and anti-EPO antibodies ELISA kits.

## Measurement of complete blood count

The samples were transported to the Postgraduate Laboratory of the University for Development Studies, Tamale Ghana, where complete blood counts were determined (on each day of sample collection). A fully-automated five-part haematology analyzer (URIT-5230, China) was used for the estimation of CBC parameters.

The automated machine works on the principles of impedance measurement (Coulter principle) and spectrophotometry for haemoglobin estimation. The blood cells are measured by creating an electric field around a calibrated micro-aperture through which the blood cells flow after being diluted in an electrolytic diluent. Haemoglobin and its derivatives: haemoglobin, methaemoglobin (Hi), and carboxyhaemoglobin (HbCO) except sulfhaemoglobin (SHb),

are converted to cyanmethaemoglobin (HiCN) by the lysing reagent, and the absorbance is measured at 540 nm and displayed on the screen of the analyzer. Anaemia in pregnancy diagnosis was based on the recommendations of the World Health Organization (WHO) (haemoglobin (Hb) concentration < 11.0 g/dL in peripheral blood) [22].

## Preparation and staining of thick and thin blood smears

The preparation, fixing and staining of blood smears was adopted from the White [4] and W. H.O, protocols [23], as described below. Thick and thin blood films were made on the same microscope slide for each participant to identify, speciate and estimate parasite density. For the thick blood film preparation, a small aliquot of blood (6 μL) was placed 10 mm from the frosted end of the slide using a micropipette and spread evenly in a circular motion to cover a diameter of 1.0–1.2 cm. A second (2 μL) of blood was then placed 1.0 cm from the thick film and spread uniformly along the length of the same slide using the edge of a second slide to make a thin film. The thin smear was fixed in absolute methanol after the slides were air-dried. The thick and thin films on each slide were then stained in a freshly-prepared 10% Giemsa stain solution using buffer (pH 7.2) for 15 minutes, and rinsed under a mild stream of water and air dried. Finally, the films were observed under the microscope using the X100 objective lens (Olympus CX 21 light microscope).

## *P. falciparum* parasite density estimation

The number of parasites was counted against approximately 200 or 500 white blood cells (WBCs), depending on the number of malaria parasites counted under the microscope, to determine parasitaemia using hand tally counters, as adopted from the W.H.O, [23]. The number of parasites per microliter of blood was then determined using the equation:

$$\text{Parasite Density (parasites/μL)} = \frac{\text{Parasite count x } 8000}{\textit{Set range of WBC}(200/500)}$$

## *P. falciparum* parasite speciation

The criteria for *P. falciparum* speciation from the thin blood film was adopted from the W. H.O guidelines for malaria diagnosis [23]. *P. falciparum* microscopically presents with trophozoites that appear ringed and comma-shaped, usually occupy about $1/5^{th}$ to $1/3^{rd}$ the diameter of the infected RBC, with fine and regular cytoplasm, marginal or accolè forms present, double chromatin dots and may contain Maurer's dots. The schizonts of *P. falciparum* are relatively small with 12–30 merozoites in a compact cluster and may contain a single dark pigment.

## Serum anti-erythropoietin and erythropoietin assay using ELISA

Serum EPO and anti-EPO were assayed by the sandwich ELISA method using commercially-prepared ELISA kits (Biobase, China). All procedures were carried out following the manufacturer's instructions for each analyte. The ELISA plates were washed using an automated ELISA washer (Poweam, WHYM201, China) while the plasma concentrations of the analytes were determined using a microplate ELISA reader (Poweam, WHYM200, China). The ELISA laboratory investigations were done at the Postgraduate Laboratory, University for Development Studies, Ghana.

### Ethical consideration and informed consent

The study strictly followed the principles enshrined in the Helsinki Declaration on research involving humans [24]. Ethical approval was sought from the Research and Ethics Committee of the University for Development Studies, Tamale, Ghana (UDS/RB/027/22). Permission was also obtained from the management of TTH. Written or oral informed consent was also obtained from the study participants before recruiting into the study.

### Statistical analysis

The data generated were analyzed with IBM Statistical Package for the Social Sciences (SPSS) software version 22.0, (IBM Corp., Armonk, NY, USA). Normality was tested with one-sample Kolmogorov-Smirnov test and Shapiro-Wilk normality tests. Parametric data and non-parametric data were presented as means ± standard deviation and medians ($25^{th}$-$75^{th}$ percentiles) respectively. Frequencies and percentages were calculated to enable comparisons of characteristics between pregnant women with *P. falciparum* malaria and normal controls. The Chi-square ($X^2$) or Fisher's exact test was used appropriately to test the descriptive statistics for the categorical variables. The Student's T test or the Mann–Whitney U test whenever applicable, were used for the continuous variables. The correlation between *P. falciparum* parasite density and serum EPO antigen levels was assessed with the Spearman rank test. Continuous variables within the three groups (Pregnant women with Pf. Malaria only, pregnant women with Pf. Malaria and anaemia, and pregnant women without malaria) were compared using the Kruskal-Wallis analysis of variance. A $p < 0.05$ was considered statistically significant.

## Results

### Demographic and clinical characteristics of the study participants

Table 1 shows the demographic and clinical characteristics of the study participants, stratified by *P. falciparum* malaria status. Of the 85 pregnant women recruited into the study, 55 (64.7%) had *P. falciparum* malaria and 30 (35.3%) had no malaria. The mean age of the study participants was 29±4.8 years with a majority (37.6%) within 25–29 years and 12 (14.2%) being either 35 years or above. The majority of the participants were multigravida women (59/69.4%), multiparous (45/51.7%), in their third trimester (39/45.9%), had more than two years inter-pregnancy intervals (53/76.8%), and 58 (80%) on iron and or folic acid supplement.

### Haemogram of the study participants stratified by the presence or absence of *P. falciparum* malaria

Table 2 shows the haemogram of the study participants stratified by the presence or absence of *P. falciparum* malaria. The median complete blood count parameters of the 85 participants were Hb, g/dL [11.6 (10.7–11.9)]; red blood cell (RBC) $x10^{12}$/L [3.7 (3.5–3.9)]; Haematocrit (HCT)% [32.5 (30.0–33.8)]; Mean Cell Haemoglobin (MCH), pg [28.9 (27.6–31.3)]; Mean Cell Haemoglobin Concentration (MCHC), g/dL [36.0 (35.0–37.0)]; Red Cell Distribution Width-Coefficient of Variation (RDW-CV)% [11.3 (10.5–12.1)]; absolute lymphocyte count $x10^9$/L [2.2 (1.8–2.9)]; absolute monocyte count $x10^9$/L [0.4 (0.3–0.6)]; absolute eosinophil count $x10^9$/L [0.09 (0.06–0.14)]; absolute basophil count$x10^9$/L [0.01 (0.01–0.02)]; Platelet count $x10^9$/L [197.0 (136.5–306.0)]; and Platelet Distribution Width (PDW)% [6.8 (6.3–7.6)]. Also, the Mean Cell Volume (MCV), Total White Blood Cell (TWBC), absolute neutrophil count and Mean Platelet Volume (MPV) were 80.2±6.9 fL, 8.0±1.9 $x10^9$/L; 5.0±1.5 $x10^9$/L; and 6.0 ±0.7 fL respectively.

**Table 1. Demographic and clinical characteristics of the study participants.**

| Variables | Total (N = 85) | Pregnant Women | | p-value |
|---|---|---|---|---|
| | | With Pf. Malaria (N = 55, 64.7%) | Without Pf. Malaria (N = 30, 35.3%) | |
| **Age (Years)** | 29.3±4.8 | 29.1±4.6 | 29.7±5.3 | 0.593 |
| **Age Category** | | | | 0.694 |
| 20–24 | 15 (17.6) | 9 (16.4) | 6 (20.0) | |
| 25–29 | 32 (37.6) | 22 (40.0) | 10 (33.3) | |
| 30–34 | 26 (30.6) | 18 (32.7) | 8 (26.7) | |
| ≥35 | 12 (14.2) | 6 (10.9) | 6 (20.0) | |
| **Religion** | | | | 0.517 |
| Islam | 71 (83.5) | 47 (85.5) | 24 (80.0) | |
| Christianity | 14 (16.5) | 8 (14.5) | 6 (20.0) | |
| **Ethnicity** | | | | 0.441 |
| Dagomba | 61 (71.8) | 41 (74.5) | 20 (66.7) | |
| Others | 24 (28.2) | 14 (25.5) | 10 (33.3) | |
| **Gravidity** | | | | 0.223 |
| Primagravida | 13 (15.3) | 7 (12.7) | 6 (20.0) | |
| Multigravida | 59 (69.4) | 38 (69.1) | 21 (70.0) | |
| Grand Multigravida | 13 (15.3) | 10 (18.2) | 3 (10.0) | |
| **Parity** | | | | 0.057 |
| Nulliparity | 19 (22.4) | 10 (18.2) | 9 (30.0) | |
| Primaparity | 22 (25.9) | 12 (21.8) | 10 (33.3) | |
| Multiparity | 45 (51.7) | 33 (60.0) | 11 (36.7) | |
| **Gestational Age** | | | | 0.970 |
| 1st Trimester | 11 (12.9) | 7 (12.7) | 4 (13.3) | |
| 2nd Trimester | 35 (41.2) | 23 (41.8) | 12 (40.0) | |
| 3rd Trimester | 39 (45.9) | 25 (45.5) | 14 (46.7) | |
| **BMI** | 27.2±5.7 | 26.5±5.2 | 28.4±6.5 | 0.156 |
| **IPI (69)** | | | | 0.936 |
| **<2 years** | 16 (23.2) | 11 (22.9) | 5 (23.8) | |
| **>2 years** | 53 (76.8) | 37 (77.1) | 16 (76.2) | |
| **IFA Supplement** | | | | 0.154 |
| **Yes** | 58 (80) | 41 (74.5) | 27 (90.0) | |
| **No** | 17 (20) | 14 (25.5) | 3 (10) | |

N = number of participants, BMI = body mass index, IPI = inter-pregnancy interval, IFA = iron/folic acid. Parametric data (presented in mean ± standard deviation) were compared by Student T-Test; Non-parametric data (presented in median ($25^{th}$-$75^{th}$ percentiles) were compared by Mann Whitney U-Test. Categorical data presented in frequencies with corresponding percentages in parenthesis. $p<0.05$ was considered significant.

Between the cases and controls, Hb (g/dL), [11.4 (9.9–11.7) vs 11.9 (11.7–12.5), $p<0.001$]; RBC x$10^{12}$/L, [3.6 (3.3–3.8) vs 3.9 (3.7–4.1), $p<0.001$]; HCT%, [32.2 (28.1–33.5) vs 32.9 (31.6–35.0), $p = 0.006$]; MCV (fL), [78.5±7.4 vs 83.3±4.4, $p = 0.002$]; MCH (pg), [28.6 (27.2–30.1) vs 31.1 (28.8–32.2), $p = 0.001$]; MCHC (g/dL), [36.0 (35.0–37.0) vs 37.0 (36.0–38.0), $p = 0.023$]; absolute lymphocyte count x$10^9$/L, [2.0 (1.7–2.7) vs 2.6 (2.0–3.1), $p = 0.007$]; Platelet x$10^9$/L, [151.0 (125.0–194.0) vs 341.0 (287.50–372.5), $p<0.001$] and PDW%, [6.8 (6.1–7.6) vs 7.4 (6.5–7.9), $p = 0.034$] were significantly lower among the pregnant women infected with *P. falciparum* compared to their counterparts without malaria. But RDW-CV, TWBC, absolute neutrophil, monocyte, eosinophil, basophil counts and MPV did not differ significantly between the two groups.

**Table 2. Haemogram of the study participants stratified by the presence or absence of *P. falciparum* malaria.**

| Variables | Total (N = 85) | Pregnant Women | | p-value |
|---|---|---|---|---|
| | | With Pf. Malaria (N = 55, 64.7%) | Without Pf. Malaria (N = 30, 35.3%) | |
| Hb (g/dL) | 11.6 (10.7–11.9) | 11.4 (9.9–11.7) | 11.9 (11.7–12.5) | <**0.001** |
| RBC x10$^{12}$/L | 3.7 (3.5–3.9) | 3.6 (3.3–3.8) | 3.9 (3.7–4.1) | <**0.001** |
| HCT% | 32.5 (30.0–33.8) | 32.2 (28.1–33.5) | 32.9 (31.6–35.0) | **0.006** |
| MCV (fL) | 80.2±6.9 | 78.5±7.4 | 83.3±4.4 | **0.002** |
| MCH (pg) | 28.9 (27.6–31.3) | 28.6 (27.2–30.1) | 31.1 (28.8–32.2) | **0.001** |
| MCHC (g/dL) | 36.0 (35.0–37.0) | 36.0 (35.0–37.0) | 37.0 (36.0–38.0) | **0.023** |
| RDW-CV% | 11.3 (10.5–12.1) | 11.4 (10.5–12.2) | 11.2 (10.3–11.7) | 0.193 |
| TWBC x10$^9$/L | 8.0±1.9 | 7.9±2.2 | 8.0±1.4 | 0.843 |
| Neut. # x10$^9$/L | 5.0±1.5 | 5.1±1.7 | 4.9±1.3 | 0.502 |
| Lymph. # x10$^9$/L | 2.2 (1.8–2.9) | 2.0 (1.7–2.7) | 2.6 (2.0–3.1) | **0.007** |
| Mon. # x10$^9$/L | 0.4 (0.3–0.6) | 0.5 (0.3–0.6) | 0.4 (0.3–0.6) | 0.432 |
| Eos. # x10$^9$/L | 0.09 (0.06–0.14) | 0.08 (0.06–0.13) | 0.10 (0.07–0.15) | 0.334 |
| Baso. # x10$^9$/L | 0.01 (0.01–0.02) | 0.01 (0.01–0.03) | 0.01 (0.01–0.03 | 0.522 |
| Platelet x10$^9$/L | 197.0 (136.5–306.0) | 151.0 (125.0–194.0) | 341.0 (287.50–372.5) | <**0.001** |
| MPV (fL) | 6.0±0.7 | 5.9±0.6 | 6.2±0.7 | 0.073 |
| PDW% | 6.8 (6.3–7.6) | 6.8 (6.1–7.6) | 7.4 (6.5–7.9) | **0.034** |

N = Number of participants, Hb = Haemoglobin concentration, RBC = Absolute red blood cell count, HCT = Haematocrit, MCV = Mean cell volume, MCH = Mean cell haemoglobin, MCHC = Mean cell haemoglobin concentration, RDW-CV = Red blood cell distribution width-coefficient of variation, TWBC = Total white blood cell count, Neut. # = Absolute neutrophil count, Lymph. # = Absolute lymphocyte count, Mon. # = Absolute monocyte count, Eos. # = Absolute eosinophil count, Baso. # = Absolute basophil count, MPV = Mean platelet volume, PDW = Platelet distribution width. Parametric data presented as mean±standard deviation was compared by Student T-Test, and Non-parametric data presented as median (25$^{th}$-75$^{th}$) were compared by Mann Whitney U-Test. *p*<0.05 was considered significant.

## Serum erythropoietin levels among pregnant women stratified by the presence or absence of *P. falciparum* malaria

Fig 1 illustrates serum erythropoietin levels among the study participants stratified by the presence or absence of *P. falciparum* malaria. The median EPO concentration of all the eighty-five pregnant women recruited into the study was 22.60 (15.83–30.45) IU/L. Serum EPO level was relatively higher among pregnant women with *P. falciparum* malaria compared to controls without malaria. But within the cases, serum EPO was significantly higher among the pregnant women with both *P. falciparum* malaria and anaemia compared to their counterparts with only malaria [35.15 (32.10–36.70) IU/L vs 23.30 (20.90–24.30) IU/L, *p*<0.001]

## Association between serum erythropoietin levels and *P. falciparum* parasite density among pregnant women

Fig 2 shows the association between serum erythropoietin levels and *P. falciparum* parasite density among pregnant women. This study observed a significantly high positive correlation between the serum EPO levels and *P. falciparum* parasite density (r = 0.978, *p* = 0.004) among pregnant women.

## Prevalence of serum anti-EPO antibodies among (a) the study participants and (b) *P. falciparum*-infected pregnant women

Fig 3 illustrates the prevalence of serum anti-EPO antibodies among the study participants. Of the 85 pregnant women in this study, the anti-EPO antibodies were detected in 5 (5.9%) of

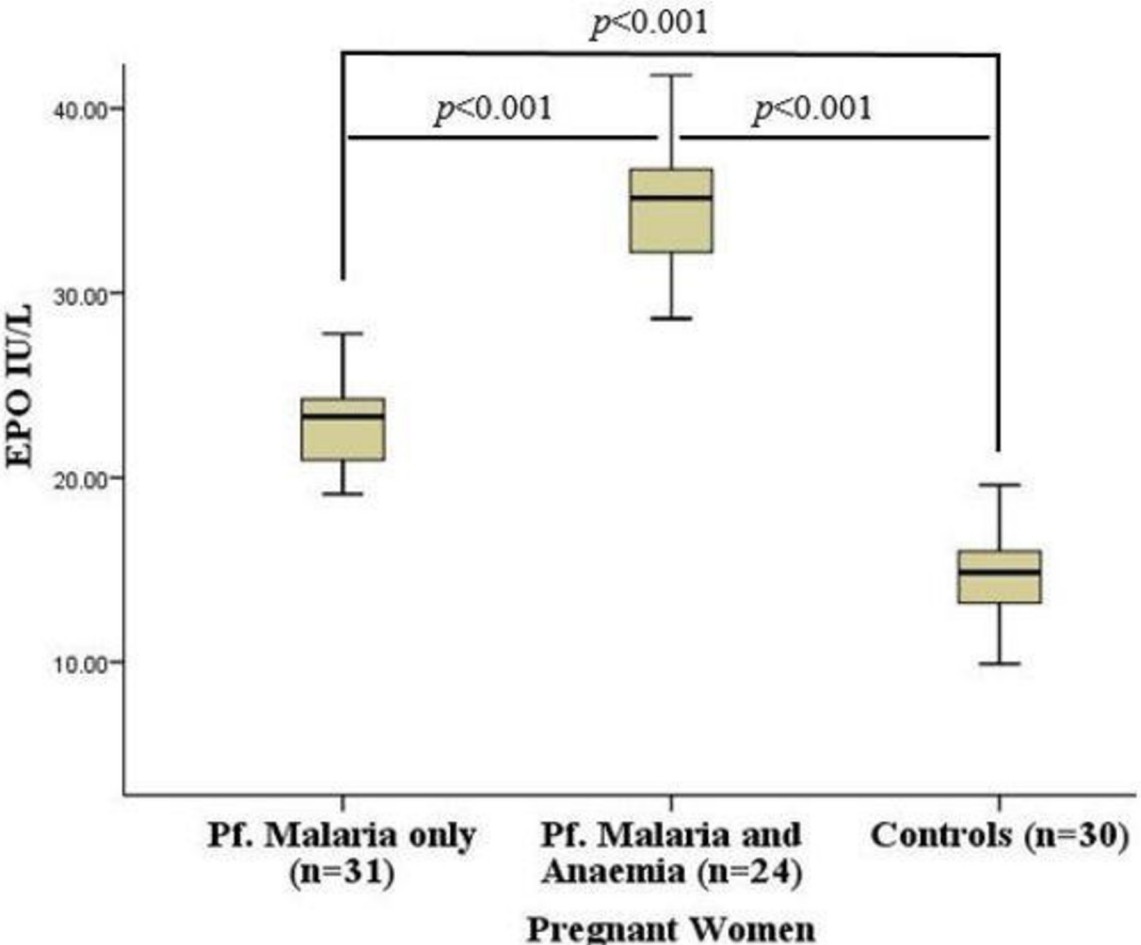

**Fig 1. Serum erythropoietin levels among pregnant women stratified by the presence or absence of *P. falciparum* malaria.**
EPO = Erythropoietin, p/μL = Parasite per Microlitre. Data were compared using Mann Whitney U-Test. $p < 0.05$ was considered significant.

them. Also, the prevalence of anti-EPO antibodies among the fifty-five *P. falciparum*-infected pregnant women was 9.1%.

### Association between serum anti-erythropoietin antibodies and *P. falciparum* parasite density among pregnant women

Fig 4 illustrates the association between serum anti-erythropoietin antibodies and *P. falciparum* parasite density among the study participants. The median *P. falciparum* parasite density was significantly higher among pregnant women with anti-EPO antibodies compared to their counterparts without the antibodies: [24019.0 parasite/μL (20640.5–28727.5) vs 2202.5 (1230.0–3629.5), $p < 0.001$].

### Haemogram of the *P. falciparum*-infected pregnant women stratified by the presence or absence of serum anti-erythropoietin antibodies

Table 3 shows the haemogram of the *P. falciparum-infected* pregnant women stratified by the presence or absence of serum anti-erythropoietin antibodies.

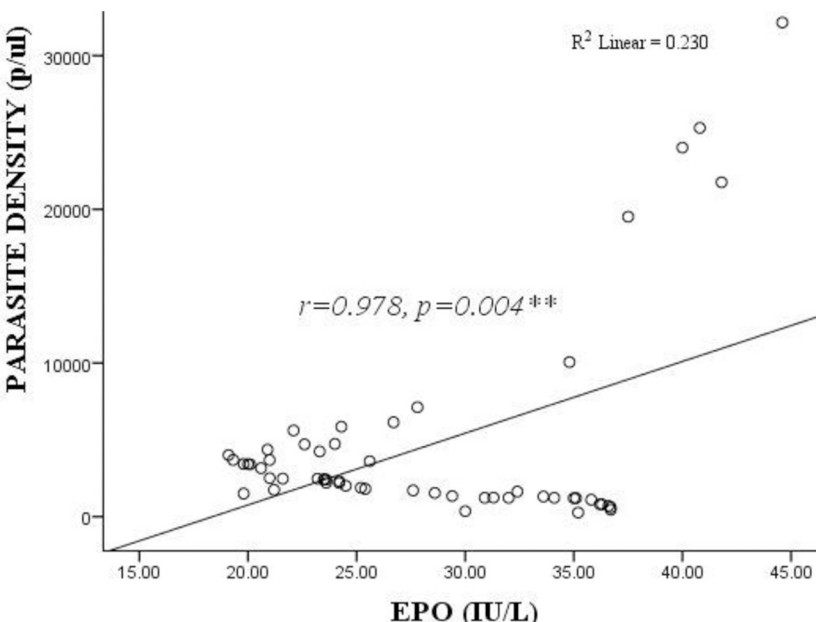

**Fig 2. Association between serum erythropoietin levels and *P. falciparum* parasite density among pregnant women.** **Correlation was significant at 0.01 level, EPO = Erythropoietin. Spearman correlation. *p*<0.05 considered statistically significant.

*P. falciparum-infected* pregnant women who had anti-EPO antibodies in their sera were relatively younger [25.0 (23.0–26.5) years] than those without the antibodies [29.0 (27.0–32.0) years], *p* = 0.013. Haemoglobin (Hb), g/dL: [7.5 (7.2–8.4) vs 29.0 (27.0–32.0), *p*<0.001]; RBC x10$^{12}$/L: [2.8 (2.3–2.9) vs 3.6 (3.5–3.8), *p*<0.001] and HCT%: [25.0 (21.9–25.4) vs 32.5 (29.4–33.5), *p*<0.001 were significantly lower among the *P. falciparum-infected* pregnant women with anti-EPO antibodies than those without the antibodies in their sera. However, leucocyte and platelet parameters were not different among the *P. falciparum-infected* pregnant women with or without the anti-EPO antibodies (Table 3).

### Age and serum erythropoietin levels and anti-erythropoietin antibody levels among the *P. falciparum-infected* pregnant women stratified by the presence or absence of anaemia

Table 4 shows age, serum erythropoietin levels and anti-erythropoietin antibody levels among the *P. falciparum*-infected pregnant women stratified by the presence or absence of anaemia.

*P. falciparum*-infected pregnant women who were anaemic were comparatively younger 27.0 years (23.0–31.5) than the non-anaemic group, 30.0 years (27.0–32.0), and this was statistically significant (*p* = 0.039). Also, *P. falciparum* malarial anaemic pregnant women had higher serum concentrations of erythropoietin compared to their non-anaemic counterparts [35.15 (32.10–36.70) IU/L vs 23.30 (20.90–24.30) IU/L, *p*<0.001]. Again, all (100%) of the pregnant women who had anti-EPO antibodies detected in their sera were anaemic, but only 38.0% of those without the anti-EPO antibodies were anaemic and this was statistically significant (*p* = 0.012) as shown in Table 4.

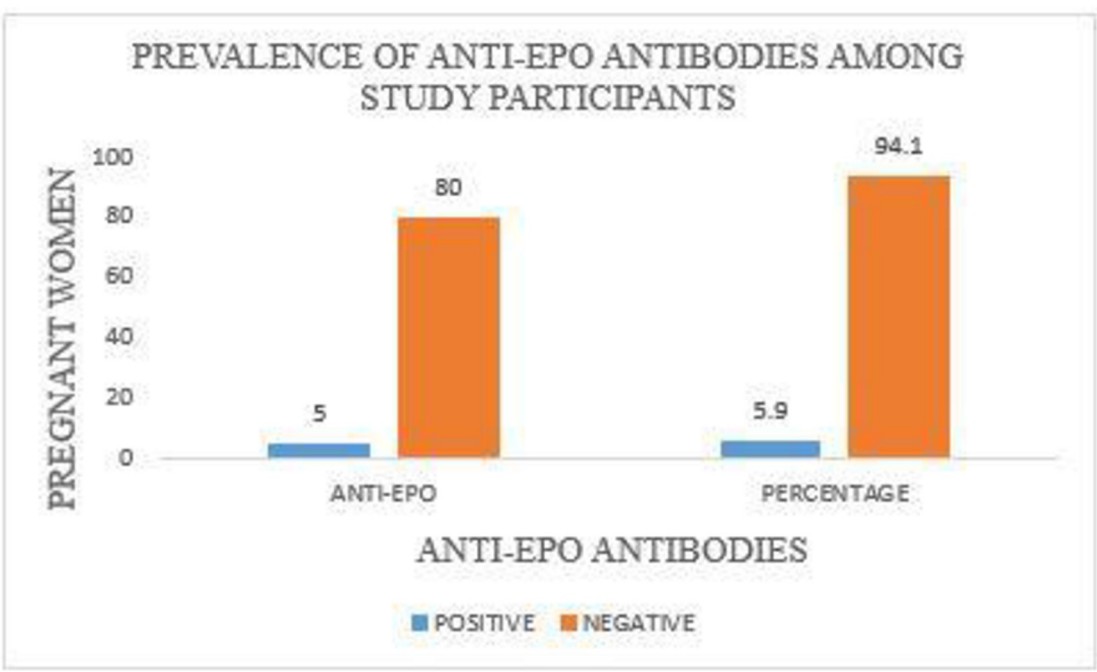

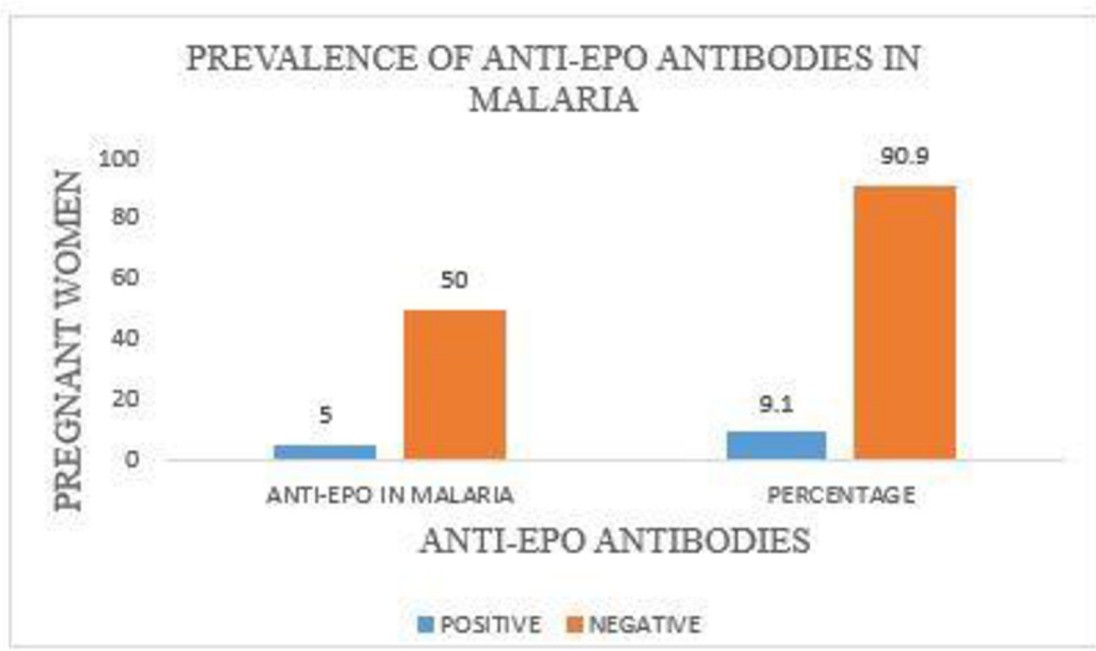

**Fig 3. Prevalence of serum anti-EPO antibodies among (a) the Study participants and (b) *P. falciparum*-infected pregnant women.** EPO = Erythropoietin.

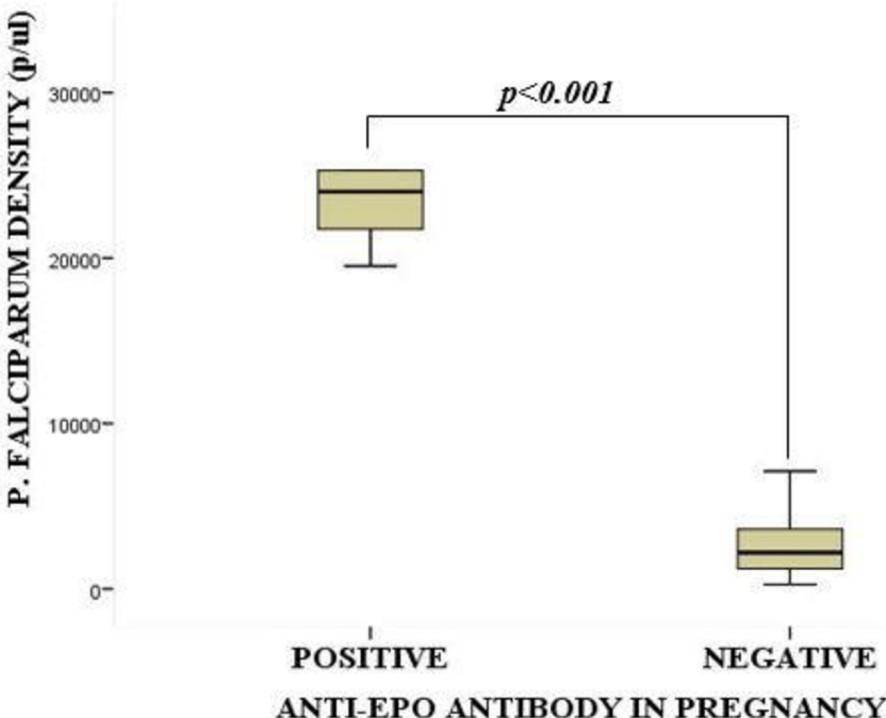

**Fig 4. Association between serum anti-erythropoietin antibodies and *P. falciparum* parasite density among pregnant women.** *Anti-EPO = Anti-Erythropoietin*, p/µL = Parasite per Microlitre. Data were compared using Mann Whitney U-Test. *p*<0.05 was considered significant.

### Serum erythropoietin levels of the *P. falciparum*-infected pregnant women stratified by the presence or absence of serum anti-Erythropoietin antibodies

The median serum erythropoietin levels of the *P. falciparum*-infected pregnant women were 22.60 (15.83–30.45) IU/L. *P. falciparum*-infected pregnant women with anti-EPO antibodies in their sera had significantly higher EPO levels compared to those without the antibodies [40.80 (38.75–43.20) vs 24.85 (21.98–32.70), *p*<0.001] as shown in Fig 5.

### Obstetric characteristics and serum erythropoietin levels among *P. falciparum*-infected pregnant women

Table 5 shows the association between obstetric characteristics and serum erythropoietin levels among the *P. falciparum*-infected pregnant women.

Pregnant women's age, gestational age, gravidity, parity, inter-pregnancy interval and being on iron and or folate did not have any association with serum EPO levels during *P. falciparum* malaria in pregnancy (Table 5).

### Linear regression analyses of obstetric characteristics associated with anti-EPO antibodies among *P. falciparum-infected* pregnant women

Table 6 shows linear regression analyses of obstetric characteristics associated with anti-EPO antibody production among *P. falciparum-infected* pregnant women. Participants' age was

**Table 3. Haemogram of the *P. falciparum*-infected pregnant women stratified by the presence or absence of serum anti-erythropoietin antibodies.**

| Variables | Pf. Infected Pregnant Women | | p-value |
|---|---|---|---|
| | Anti-EPO Present (N = 5) | Anti-EPO Absent (N = 50) | |
| Age (years) | 25.0 (23.0–26.5) | 29.0 (27.0–32.0) | **0.013** |
| Hb (g/dL) | 7.5 (7.2–8.4) | 11.5 (10.4–11.7) | **<0.001** |
| RBC x10$^{12}$/L | 2.8 (2.3–2.9) | 3.6 (3.5–3.8) | **<0.001** |
| HCT% | 25.0 (21.9–25.4) | 32.5 (29.4–33.5) | **<0.001** |
| MCV (fL) | 81.1±9.6 | 78.2±7.2 | 0.483 |
| MCH (pg) | 30.0 (25.9–33.2) | 28.6 (27.1–29.7) | 0.366 |
| MCHC (g/dL) | 37.0 (35.5–37.5) | 36.0 (34.7–37.0) | 0.390 |
| RDW-CV% | 11.3 (10.6–11.8) | 11.5 (10.5–12.2) | 0.631 |
| TWBC x10$^9$/L | 8.2±2.6 | 7.9±2.2 | 0.759 |
| Neut. # x10$^9$/L | 5.3±1.6 | 5.1±1.7 | 0.691 |
| Lymph. # x10$^9$/L | 2.3 (1.3–3.2) | 2.0 (1.7–2.7) | 0.977 |
| Mon. # x10$^9$/L | 0.6 (0.3–0.7) | 0.5 (0.3–0.6) | 0.413 |
| Eos. # x10$^9$/L | 0.07 (0.05–0.10) | 0.08 (0.06–0.13) | 0.489 |
| Baso. # x10$^9$/L | 0.01 (0.01–0.02) | 0.01 (0.01–0.02) | 0.218 |
| Platelet x10$^9$/L | 131.0 (118.0–141.0) | 162.0 (125.8–197.3) | 0.050 |
| MPV (fL) | 6.2±0.6 | 5.9±0.6 | 0.255 |
| PDW% | 7.2 (6.8–8.9) | 6.8 (6.1–7.6) | 0.103 |

N = Number of participants, Hb = Haemoglobin concentration, RBC = Absolute red blood cell count, HCT = Haematocrit, MCV = Mean cell volume, MCH = Mean cell haemoglobin, MCHC = Mean cell haemoglobin concentration, RDW-CV = Red blood cell distribution width-coefficient of variation, TWBC = Total white blood cell count, Neut. # = Absolute neutrophil count, Lymph. # = Absolute lymphocyte count, Mon. # = Absolute monocyte count, Eos. # = Absolute eosinophil count, Baso. # = Absolute basophil count, MPV = Mean platelet volume, PDW = Platelet distribution width. Parametric data presented as mean±standard deviation was compared by Student T-Test and Non-parametric data presented as median (25$^{th}$-75$^{th}$) were compared by Mann Whitney U-Test. $p<0.05$ was considered significant.

significantly associated with serum anti-EPO antibodies in *P. falciparum* malaria in pregnancy ($p = 0.019$). However, gestational age, parity, gravidity, inter-pregnancy interval (IPI) and being on iron and or folate supplements had no association with serum anti-EPO among pregnant women with *P. falciparum* malaria.

**Table 4. Age, serum erythropoietin levels and anti-erythropoietin antibodies levels among the *P. falciparum*-infected pregnant women stratified by the presence or absence of anaemia.**

| Variables | Anaemic Status of Pf. Infected Pregnant Women | | p-value |
|---|---|---|---|
| | Anaemic (N = 24) | Non-Anaemic (N = 31) | |
| **Age (years)** | 27.0 (23.0–31.5) | 30.0 (27.0–32.0) | **0.039** |
| **EPO Conc. (IU/L)** | 35.15 (32.10–36.70) | 23.30 (20.90–24.30) | **<0.001** |
| **Anti-EPO Status** | | | **0.012** |
| Positive | 5 (100) | 0 | |
| Negative | 19 (38.0%) | 31 (62.0%) | |
| **Anti-EPO Antibody Conc. (IU/L)** | 7.26 (6.18–10.25) | - | - |

N = number of participants, EPO = Erythropoietin, IU/L = International Unit per Litre. Numeric Data were compared using Mann Whitney and Fisher's Exact Test for Anti-EPO status. $p<0.05$ was considered significant.

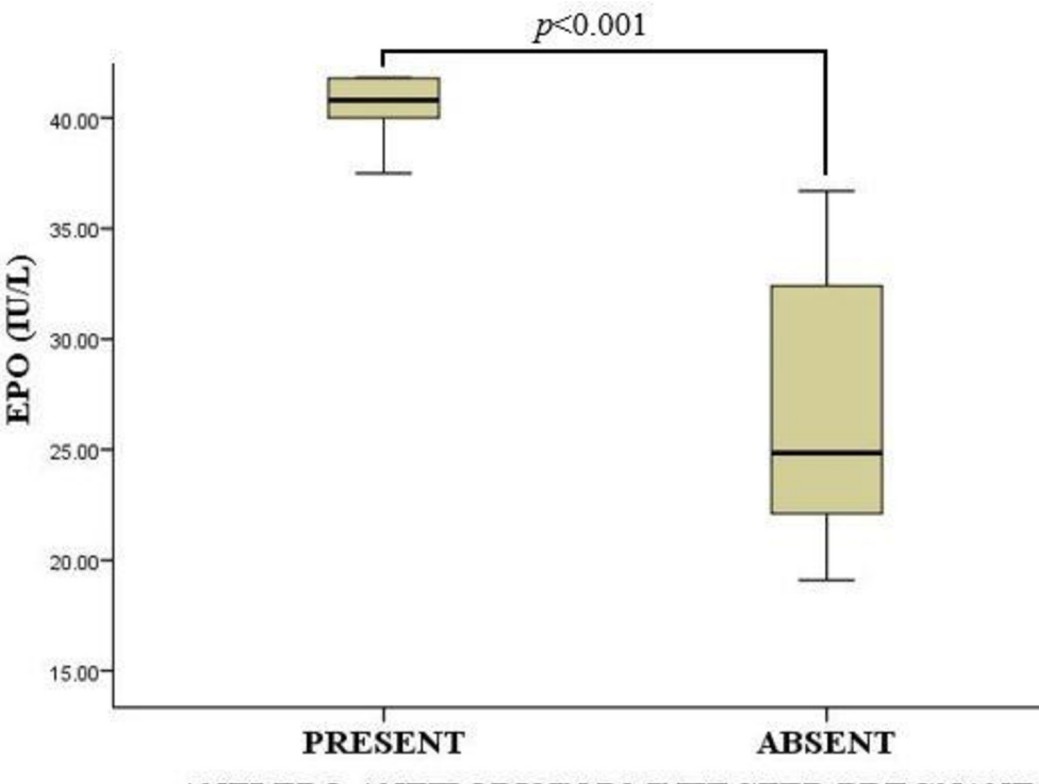

**Fig 5. Serum erythropoietin levels of the *P. falciparum*-infected pregnant women stratified by the presence or absence of serum anti-erythropoietin antibodies.** EPO = Erythropoietin; IU/L = International Unit per Litre. Data were compared using Mann Whitney U-Test. $p < 0.05$ was considered significant.

## Discussion

*P. falciparum* malaria remains the major contributor to the development of anaemia in pregnancy in Ghana [6]. Malarial anaemia occurs through complex mechanisms and presents with life-threatening complications to both maternal and foetal lives [4, 8]. The presence of anti-EPO antibodies may be related to the development of anaemia in disease conditions such as malaria [25], HIV/AIDS and systemic lupus erythematosus (SLE) [19]. This study assessed the presence of anti-EPO antibodies and their association with anaemia in *Plasmodium falciparum*-infected pregnant women.

Haemoglobin, RBC, HCT and platelet were significantly lower among pregnant women infected with *P. falciparum*. About 6.0% out of the 85 total participants had anti-EPO antibodies in their sera, and the prevalence of anti-EPO antibodies among the *P. falciparum*-infected pregnant women was 9.1%. *P. falciparum-infected* pregnant women with anti-EPO antibodies had lower Hb, RBC, and HCT, but higher EPO levels compared with those without the antibodies. Younger age and high parasite density are significantly associated with the presence of *P. falciparum*-related anti-EPO antibodies in pregnancy. Also, younger age and the presence of anti-EPO antibodies in sera related to the development of anaemia in pregnant women with *P. falciparum* malaria.

The significantly lower red cell indices observed in pregnant women with *P. falciparum* malaria are consistent with other studies conducted in Nigeria [26, 27] and Ghana [28]. *P.*

**Table 5. Obstetric characteristics and serum erythropoietin levels among *P. falciparum*-infected pregnant women.**

| Variables | EPO Levels (IU/L) | 95%CI | S.E. | p-value |
|---|---|---|---|---|
| **Age Category** | | | | 0.791 |
| 20–24 | 29.4 (14.7–32.8) | 16.9–33.8 | 3.7 | |
| 25–29 | 22.6 (18.2–31.6) | 18.7–27.3 | 2.1 | |
| 30–34 | 25.0 (19.1–32.0) | 22.2–29.1 | 1.6 | |
| >35 | 23.4 (17.5–27.2) | 16.3–30.9 | 2.8 | |
| **Gestational Age** | | | | 0.260 |
| 1st Trimester | 29.4 (20.6–33.6) | 17.3–35.9 | 3.8 | |
| 2nd Trimester | 21.0 (17.6–27.6) | 18.6–25.9 | 1.8 | |
| 3rd Trimester | 24.2 (19.9–34.0) | 22.1–29.1 | 1.7 | |
| **Gravidity** | | | | 0.899 |
| Primagravida | 24.4 (17.7–32.4) | 16.6–22.1 | 3.4 | |
| Multigravida | 23.6 (19.0–29.9) | 21.3–27.0 | 1.4 | |
| Grand Multigravida | 23.9 (18.7–32.3) | 18.5–31.1 | 2.8 | |
| **Parity** | | | | 0.299 |
| Nulliparity | 23.6 (14.8–31.3) | 16.1–29.9 | 3.09 | |
| Primaparity | 28.5 (20.8–33.3) | 23.6–31.7 | 1.8 | |
| Multiparity | 23.3 (18.0–30.3) | 20.3–26.7 | 1.6 | |
| **IFA Supplement** | | | | 0.493 |
| Yes | 24.2 (19.1–29.9) | 22.5–27.7 | 1.3 | |
| No | 20.6 (17.2–34.7) | 17.4–30.5 | 3.0 | |
| **IP** | | | | 0.825 |
| <2 years | 20.6 (14.7–36.6) | 16.1–31.5 | 3.5 | |
| >2 years | 23.8 (19.4–31.6) | 22.5–27.6 | 1.2 | |

EPO = Erythropoietin, IU/L = International Unit per Litre, CL = Confidence Interval, S.E. = Standard Error. Bivariate data were generated with Mann Whitney U-Test, Multivariate by Kruskal-Wallis Test. *p*<0.05 was considered significant.

*falciparum* malaria-related anaemia develops through complex mechanisms including the suppression of erythropoiesis, placental sequestration of infected erythrocytes, and haemolysis of both infected and non-infected erythrocytes, inflammatory mediators such as TNF-alpha, IL-1 and IL-6 and direct interaction with parasite variant surface antigens [4, 10].

Thrombocytopaenia during *P. falciparum* malaria has been reported earlier in Thailand [29] and Ghana [28]. The thrombocytopaenia was thought to be due to the associated splenic

**Table 6. Linear regression analyses of obstetric characteristics associated with anti-EPO antibodies among *P. falciparum-infected* pregnant women.**

| Variables | Beta | S.E. | 95% CI | p-value |
|---|---|---|---|---|
| **Age** | 0.381 | 0.010 | 0.004–0.044 | **0.019** |
| **Gestational Age** | -0.043 | 0.005 | -0.012–0.008 | 0.754 |
| **Parity** | -0.283 | 0.053 | -0.172–0.042 | 0.229 |
| **Gravidity** | 0.131 | 0.044 | -0.063–0.113 | 0.576 |
| **IPI** | -0.217 | 0.085 | -0.295–0.046 | 0.148 |
| **IFA Supplement** | -0.0159 | 0.080 | -0.247–0.075 | **0.287** |

IPI = Inter-Pregnancy Interval, IFA = Iron/Folic Acid. CL = Confidence Interval, S.E. = Standard Error. *p*<0.05 was considered significant.

sequestration of thrombocytes, as well as the reduced bone marrow haemopoietic activities [30] during *P. falciparum* infestation. Findings from our current study agree with the earlier findings [28–30].

The prevalence of anti-EPO antibodies in the study participants and among the pregnant women infected with *P. falciparum* were slightly higher compared to a recent study in the middle belt of Ghana which recorded a prevalence of 3.5% in the entire study participants and 2.3% in the *P. falciparum-infected* children [20]. The differences in the findings may be due to the variations in the study participants; whilst this study recruited pregnant women, the Addai-Mensah et al. [20] study involved only children aged 1–10 years. Again, anti-EPO antibodies have been identified in HIV/AIDS and autoimmune diseases such as systemic lupus erythematosus [19]. External factors including *P. falciparum* malaria may induce a pro-inflammatory environment that promotes the activation of autoreactive lymphocytes, producing antibodies such as anti-EPO [9, 11].

The higher *P. falciparum* parasite density identified among the pregnant women with anti-EPO antibodies in this study suggests a possible contribution of parasitaemia to the development of the antibody in malaria. In severe *P. falciparum* malaria with corresponding high parasite density [29, 31], a more intense pro-inflammatory environment with an elevation of cytokines including TNF-alpha, IL-1 and IL-6 may be created and this could contribute to the delayed erythrocytic response seen in *P. falciparum* induced anaemia. Also, the pro-inflammatory mediators may induce the production of antibodies [9, 11] such as anti-EPO antibodies as may occur in *P. falciparum* malaria, especially during pregnancy where the immune response is hyperactive. Thus, high *P. falciparum* parasitaemia could stimulate the release of anti-EPO antibodies in pregnancy.

In this study, pregnant women with *P. falciparum* malaria who were positive for serum anti-EPO antibodies were significantly younger; younger age may therefore stimulate the production of anti-EPO in pregnant women with *P. falciparum*, and this is consistent with a previous study on patients with systemic lupus erythematosus, an autoimmune disease [32]. The biological and clinical significance of this finding remains unclear, but may be related to the enhanced ability of the immune system to mount an immune response at a younger age [33, 34]. The presence of serum anti-EPO antibodies in the *P. falciparum*-infected pregnant women significantly affected peripheral erythrocytes, haemoglobin and haematocrit in this study. The relatively lower Hb, RBC count and haematocrit observed among the *P. falciparum* infected pregnant women with anti-EPO antibodies in peripheral blood may be due to a possible neutralizing effect of the antibodies on erythropoietin, which eventually downregulates erythropoiesis. Interestingly, all the malaria-infected pregnant women with serum anti-EPO antibodies in this study were anaemic, signifying the antibodies' association with *P. falciparum* malaria-related anaemia in pregnancy. A similar finding was observed in a previous study involving patients suffering from systemic lupus erythematosus [35], which suggested significant association between anti-EPO antibodies and anaemia. Further studies to investigate the neutralizing effect anti-EPO antibodies could have on EPO in the development of anaemia would be essential. This finding in the current study, however, contradicts a recent study conducted on children in Ghana by Addai-Mensah et al. [20] which found no association between serum anti-EPO antibodies and anaemia in *P. falciparum* malaria. The discrepancy could be attributed to the differences in the study population. Whiles the present study recruited pregnant women in northern Ghana, the Addai-Mensah et al. [20] study involved children from 1–10 years within the middle belt of the country.

Again, the younger age of the *P. falciparum*-infected pregnant women was significantly associated with the development of anaemia, and this is consistent with previous studies in South Africa [36], Pakistan [37] and Nigeria [38]. The relationship between younger age and

malaria-related anaemia in pregnancy is not fully understood, but could be attributed to the fact that the development of immunity to malaria, for instance, antibodies that inhibit the adherence of parasites to chondroitin sulphate-A (CS-A) in the placenta, is acquired in later years and with subsequent pregnancies [39, 40].

Apart from age that largely associated with the presence of anti-EPO antibodies in *P. falciparum* malaria in pregnancy, other obstetric characteristics such as gravidity, parity, gestational age, inter-pregnancy interval and iron and or folic acid supplementation had no association with the antibodies' presence in the serum.

This study was limited by our inability to assess the neutralizing effect of anti-EPO on EPO. The study could not also ascertain the influence of anti-EPO antibodies on EPO receptors.

## Conclusion

The prevalence of anti-EPO antibodies among pregnant women with *P. falciparum* malaria was high. *P. falciparum* parasite density and younger age could stimulate the production of anti-EPO antibodies, and the antibodies may contribute to the development of malaria-induced anaemia in pregnancy. Screening for anti-EPO antibodies should be considered in pregnant women with *P. falciparum* malaria. Further studies should be conducted to elucidate the significance and effect of anti-EPO antibodies on the health of pregnant women, especially those presenting with *P. falciparum* malaria and anaemia.

## Acknowledgments

Authors are grateful for the enormous contributions of senior members of the Department of Biomedical Laboratory Sciences and the Department of Haematology, School of Allied Health Sciences, University for Development Studies, Tamale, Ghana. We appreciate the support of the staff of the Clinical Laboratory Department, Tamale Teaching Hospital, Ghana, for their immense support. A big thank you to all pregnant women who willingly involved themselves in the study.

## Author Contributions

**Conceptualization:** Charles Nkansah, Simon Bannison Bani, Kofi Mensah, Samuel Kwasi Appiah, Eugene Mensah Agyare, Peace Esenam Agbadza.

**Data curation:** Eugene Mensah Agyare, Peace Esenam Agbadza, Yeduah Quansah, Rahama Issah, Samuel Yennuloom Dindiok.

**Formal analysis:** Charles Nkansah, Kofi Mensah, Felix Osei-Boakye, Eugene Mensah Agyare, Peace Esenam Agbadza.

**Investigation:** Charles Nkansah, Samuel Kwasi Appiah, Eugene Mensah Agyare, Peace Esenam Agbadza.

**Methodology:** Charles Nkansah, Eugene Mensah Agyare.

**Resources:** Eugene Mensah Agyare, Peace Esenam Agbadza, Rahama Issah, Samuel Yennuloom Dindiok.

**Supervision:** Charles Nkansah.

**Validation:** Charles Nkansah, Simon Bannison Bani, Kofi Mensah, Samuel Kwasi Appiah, Felix Osei-Boakye, Gabriel Abbam, Samira Daud, Charles Angnataa Derigubah, Dorcas Serwaa, Francis Atoroba Apodola, Felix Ejike Chukwurah.

**Visualization:** Charles Nkansah.

**Writing – original draft:** Charles Nkansah, Simon Bannison Bani, Kofi Mensah, Samuel Kwasi Appiah, Felix Osei-Boakye, Gabriel Abbam, Samira Daud, Eugene Mensah Agyare, Peace Esenam Agbadza, Charles Angnataa Derigubah, Dorcas Serwaa, Francis Atoroba Apodola, Yeduah Quansah, Rahama Issah, Samuel Yennuloom Dindiok, Felix Ejike Chukwurah.

**Writing – review & editing:** Charles Nkansah, Simon Bannison Bani, Kofi Mensah, Samuel Kwasi Appiah, Felix Osei-Boakye, Gabriel Abbam, Samira Daud, Eugene Mensah Agyare, Peace Esenam Agbadza, Charles Angnataa Derigubah, Dorcas Serwaa, Francis Atoroba Apodola, Yeduah Quansah, Rahama Issah, Samuel Yennuloom Dindiok, Felix Ejike Chukwurah.

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
