## [Decision Letter · Decision Letter 0]

3 Jan 2023

PONE-D-22-28777

Serum anti-erythropoietin antibodies among pregnant women with Plasmodium falciparum malaria and anaemia: A case-control study in northern Ghana

PLOS ONE

Dear Dr. Nkansah,

Thank you for submitting your manuscript to PLOS ONE. After careful consideration, we feel that it has merit but does not fully meet PLOS ONE’s publication criteria as it currently stands. Therefore, we invite you to submit a revised version of the manuscript that addresses the points raised during the review process.

The comments of two expert reviewers that assessed your manuscript are appended below. Both reviewers raise important concerns about some aspects of the methodology and presentation, which you should address carefully in your revised manuscript.  Note the detailed presentation of your sample size calculation is accepted as it will enhance reproducibility. Kindly proofread the manuscript and make suggested edits to improve flow.

We look forward to receiving your revised manuscript.

Kind regards,

Dr. Felix C. C. Wekere

Academic Editor

PLOS ONE

Journal Requirements:

Additional Editor Comments:

Kindly add the city and country of the manufacturer of the statistical software you used for analysis.

Also, give the full meaning of the abbreviations at first use. 

Reviewers' comments:

Reviewer's Responses to Questions

**Comments to the Author**

1. Is the manuscript technically sound, and do the data support the conclusions?

Reviewer #1: Yes

Reviewer #2: Yes

2. Has the statistical analysis been performed appropriately and rigorously? 

Reviewer #1: Yes

Reviewer #2: Yes

3. Have the authors made all data underlying the findings in their manuscript fully available?

Reviewer #1: No

Reviewer #2: Yes

4. Is the manuscript presented in an intelligible fashion and written in standard English?

Reviewer #1: Yes

Reviewer #2: Yes

5. Review Comments to the Author

Reviewer #1: the current study ascertained the prevalence of anti-EPO antibodies production and evaluated the antibodies’ relationship with Plasmodium falciparum malaria and malaria-related anaemia in pregnancy. this is a well written manuscript, the following suggestions/ questions will enhanced readers understandng

1. authors should state categorically how Plasmodium falciparum density and age relates to anti-EPO do anti-EPO enhanced or inhibit anti-EPO

2. Line 59 sentence .......... in both foetal and or maternal lives should be revised/rephrased to enhanced clarity

3.Line 100 define EPO first time usage

4. Line 74-76 ........but suggested via multiple mechanisms including suppression .....revise to enhance clarity of understanding

5. Sample size calculation as demonstrated in the manuscript is too detailed, authors should just state the name of the formulae, the prevalence used, referenced the articled from which it was sourced from and finally the sample size itself.

6. how did authors ensured that exclusion criteria were met ie P. falciparum-infected pregnant women with

comorbidities (such as infections and other haematological disorders) were excluded.

7. line 100 subheading assessment of CBC should be changed to Measurement of CBC

8. Authors failed to demonstrate how they speciated P. falciparum from other Spp. of anopheles

9. Authors should move the limitation of the study from conclusion section to the discussion section

Reviewer #2: The animal studies in the narration may not directly substitute for pregnant humans. The findings with animal studies that were not pregnant is far too wide for comparative analysis. The study is on human pregnant conditions, P. Falciparum and presence of antiEOP antibodies. It may not be absolutely necessary in the discussion page of this study.

6. PLOS authors have the option to publish the peer review history of their article (what does this mean?). If published, this will include your full peer review and any attached files.

Reviewer #1: No

Reviewer #2: **Yes: **Matthew Anyanwu

---

## [Author Response · Author response to Decision Letter 0]

10 Jan 2023

Dear Editor,

Thank you for your email. I am pleased to resubmit manuscript titled “Serum anti-erythropoietin antibodies among pregnant women with Plasmodium falciparum malaria and anaemia: A case-control study in northern Ghana “for your consideration.

Authors do appreciate your endless efforts to ensure improvement in the manuscript.

The concerns raised by the editorial team have been addressed and highlighted below and marked in the respective sections in the manuscript. I look forward to your favourable response.

Thank you.

The manuscript was edited for language usage, spelling and grammar by Grammarly Inc. USA.

General Comments

1. Initial incorrect spelling of the middle name of the last author 

Response: Authors middle name is correctly spelt: Ejike. This makes the full name Felix Ejike Chukwurah 

2. Kindly add the city and country of the manufacturer of the statistical software you used for analysis.

Response: This has been considered and inputs made

3. Also, give the full meaning of the abbreviations at first use. 

Response: This has been considered throughout the document

Reviewer #1: 

1. authors should state categorically how Plasmodium falciparum density and age relates to anti-EPO do anti-EPO enhanced or inhibit anti-EPO

Response: P. falciparum density and younger age may stimulate the production of anti-EPO in pregnant women with malaria 

2. Line 59 sentence .......... in both foetal and or maternal lives should be revised/rephrased to enhanced clarity

Response: This has been revised accordingly

3. Line 100 define EPO first time usage 

Response: EPO is first defined in line 37 of the manuscript.

4. Line 74-76 ........but suggested via multiple mechanisms including suppression .....revise to enhance clarity of understanding.

Response: This has been revised to ensure clarity

5. Sample size calculation as demonstrated in the manuscript is too detailed, authors should just state the name of the formulae, the prevalence used, referenced the articled from which it was sourced from and finally the sample size itself.

Response: The detailed presentation of the sample size determination will allow for reproducibility. 

6. how did authors ensured that exclusion criteria were met ie P. falciparum-infected pregnant women with comorbidities (such as infections and other haematological disorders) were excluded.

Response: Clinical characteristics of the pregnant women (participants) were obtained from the ante-natal care (ANC) register, which contained results on HIV, HBsAg, HCV, Syphilis, Sickling test, Hb phenotype, G6PD test, routine examinations of stool and urine, etc. Pregnant women with comorbidities were excluded from the study, as shown in lines 149-150.

7. line 100 subheading assessment of CBC should be changed to Measurement of CBC

Response: The subheading has been revised to read ‘Measurement of complete blood count’, and seen in line 165

8. Authors failed to demonstrate how they speciated P. falciparum from other Spp. of anopheles 

Response: This has been addressed and inserted in lines 198 to 205

9. Authors should move the limitation of the study from conclusion section to the discussion section

Response: The limitation of the study has been moved from the conclusion to the discussion section as recommended.

Reviewer #2: 

The animal studies in the narration may not directly substitute for pregnant humans. The findings with animal studies that were not pregnant is far too wide for comparative analysis. The study is on human pregnant conditions, P. Falciparum and presence of anti-EOP antibodies. It may not be absolutely necessary in the discussion page of this study.

Response: This has been considered and revision made

---

## [Decision Letter · Decision Letter 1]

6 Mar 2023

PONE-D-22-28777R1Serum anti-erythropoietin antibodies among pregnant women with Plasmodium falciparum malaria and anaemia: A case-control study in northern GhanaPLOS ONE

Dear Dr. Nkansah,

Thank you for submitting your manuscript to PLOS ONE. After careful consideration, we feel that it has merit but does not fully meet PLOS ONE’s publication criteria as it currently stands. Therefore, we invite you to submit a revised version of the manuscript that addresses the points raised during the review process.

Your manuscript has been assessed by an expert reviewer, whose comments are appended below. The reviewer raised important concerns about some aspects of the Methodology and  presentation, which you should address carefully in your revised manuscript. Kindly address each point carefully in your response to the reviewers’ document and revise the manuscript accordingly.

We look forward to receiving your revised manuscript.

Kind regards,

Felix Chikaike Clement Wekere

Academic Editor

PLOS ONE

Journal Requirements:

Additional Editor Comments:

Kindly  revise line 94 in the introduction. Suggestion: Delete the sentence completely as you did in the discussion section (i.e comparison with animal-mice), since your work is on humans it won't be appropriate referring to an animal study.

In Table 1, remove the word 'years' after the different ages groups since you have written it in bracket in the preceding row. In Table 5 format as suggested above for Table 1.

Reviewers' comments:

Reviewer's Responses to Questions

**Comments to the Author**

1. If the authors have adequately addressed your comments raised in a previous round of review and you feel that this manuscript is now acceptable for publication, you may indicate that here to bypass the “Comments to the Author” section, enter your conflict of interest statement in the “Confidential to Editor” section, and submit your "Accept" recommendation.

Reviewer #2: (No Response)

2. Is the manuscript technically sound, and do the data support the conclusions?

Reviewer #2: Yes

3. Has the statistical analysis been performed appropriately and rigorously? 

Reviewer #2: Yes

4. Have the authors made all data underlying the findings in their manuscript fully available?

Reviewer #2: Yes

5. Is the manuscript presented in an intelligible fashion and written in standard English?

Reviewer #2: No

6. Review Comments to the Author

Reviewer #2: Lines 419, 420, 421: The animal studies in the narration may not directly substitute for pregnant humans. The findings with animal studies that were not pregnant is far too wide for comparative analysis. The study is on human pregnant conditions, P. Falciparum and presence of antiEOP antibodies. It may not be absolutely necessary in the discussion page of this study. Similarly the study on children 0-10 years was conducted in Ghana which may be appropriate to mention but of no value to the current study. I observed also that few words of study limitations were written at the conclusion paragraph. May i suggest that it should come after the discussion paragraphs.

7. PLOS authors have the option to publish the peer review history of their article (what does this mean?). If published, this will include your full peer review and any attached files.

Reviewer #2: **Yes: **Matthew Anyanwu

---

## [Author Response · Author response to Decision Letter 1]

7 Mar 2023

Dear Editor,

Thank you for your email. I am pleased to resubmit manuscript titled “Serum anti-erythropoietin antibodies among pregnant women with Plasmodium falciparum malaria and anaemia: A case-control study in northern Ghana “for your consideration.

Authors do appreciate your endless efforts to ensure improvement in the manuscript.

The concerns raised by the editorial team have been addressed and highlighted below and marked in the respective sections in the manuscript. I look forward to your favourable response.

Thank you.

The manuscript was edited for language usage, spelling and grammar by Grammarly Inc. USA.

General Comments

1. Kindly revise line 94 in the introduction. Suggestion: Delete the sentence completely as you did in the discussion section (i.e comparison with animal-mice), since your work is on humans it won't be appropriate referring to an animal study.

Response: The entire sentence has been deleted as suggested

2. In Table 1, remove the word 'years' after the different ages groups since you have written it in bracket in the preceding row. In Table 5 format as suggested above for Table 1.Initial incorrect spelling of the middle name of the last author 

Response: The word ‘years’ has been deleted from both Tables 1 and 5 accordingly. 

Reviewer #2: 

The animal studies in the narration may not directly substitute for pregnant humans. The findings with animal studies that were not pregnant is far too wide for comparative analysis. The study is on human pregnant conditions, P. Falciparum and presence of anti-EOP antibodies. It may not be absolutely necessary in the discussion page of this study.

Response: This has been considered and revision made

---

## [Editor Report · Decision Letter 2]

8 Mar 2023

Serum anti-erythropoietin antibodies among pregnant women with Plasmodium falciparum malaria and anaemia: A case-control study in northern Ghana

PONE-D-22-28777R2

Dear Dr. Nkansah,

We’re pleased to inform you that your manuscript has been judged scientifically suitable for publication and will be formally accepted for publication once it meets all outstanding technical requirements.

Kind regards,

Felix Chikaike Clement Wekere

Academic Editor

PLOS ONE
---

## [Editor Report · Acceptance letter]

21 Mar 2023

PONE-D-22-28777R2 

*Serum anti-erythropoietin antibodies among pregnant women with Plasmodium falciparum malaria and anaemia: A case-control study in northern Ghana*

Dear Dr. Nkansah:

I'm pleased to inform you that your manuscript has been deemed suitable for publication in PLOS ONE. Congratulations! Your manuscript is now with our production department. 

Kind regards, 

on behalf of

Dr. Felix Chikaike Clement Wekere 

Academic Editor

PLOS ONE